# Remote Work Efficiency from the Employers' Perspective—What's Next?

**Zenon Pokojski, Agnieszka Kister and Marcin Lipowski ***

Faculty of Economics, Maria Curie-Sklodowskiej University, 20-031 Lublin, Poland;
zenon.pokojski@mail.umcs (Z.P.); agnieszka.kister@mail.umcs (A.K.)
* Correspondence: marcin.lipowski@mail.umcs.pl

**Abstract:** Remote work has been of interest to managers since the implementation of new information and communication technologies (ICTs). During the initial period, it was treated as an employee's privilege or even a luxury and as such it was not a popular practice. The COVID-19 pandemic and the intervening period have changed attitudes toward remote work, as it became a necessity for many organisations. However, in connection with its use, many new, previously unknown problems have arisen, such as: the organisation of remote work, the supervision and monitoring of work performance, and employee support. The present research was conducted using a standardised questionnaire computer-assisted telephone interview (CATI) method in May–June 2021 on a population of 248 enterprises, divided into micro, small, medium-sized and large entities. The research data were collected during the COVID-19 pandemic which, on the one hand, provided an exceptional opportunity to fill in the theoretical gaps that were existing in this field; however, on the other hand, it could be burdened with certain flaws due to the context of the pandemic. An enterprise's attitude to remote work has a positive influence on the efficiency of the remote work, the control of the remote work and the remote work support, with the strongest impact exerted on the last of the factors mentioned. A better attitude to remote work influences, to the largest degree, an enterprise's support for performing work from remote locations outside of corporate offices. Among the enterprises that were surveyed, the following were most frequently indicated as elements of such support: additional office equipment provided to an employee, remote work training, and the installation of additional computer programs. Financial support was declared by about 11% of the enterprises and it usually took the form of a remote work allowance or funds to cover the costs of purchasing equipment or paying for the Internet.

**Keywords:** remote work; effectiveness of remote work; support for remote work; remote work control

## 1. Introduction

Since the implementation of new ICTs, remote work has been of interest to managers. This kind of work is sometimes described as work from home, work from anywhere (WFA), telecommuting, virtual work, mobile work or flexible work [1–5]. Remote work (RW) is defined as " . . . a flexible working arrangement that allows an employee to work from a remote location outside of corporate offices or production facilities, without having personal contact with his/her co-workers but with an ability to communicate with them by means of information and communication technologies" [6]. Researchers argue that flexible work arrangements, including remote work, will continue into the future despite having many disadvantages [5]. Remote work is beneficial to employees and can have positive and negative effects on an individual level [7]. Research on remote work is particularly relevant to the COVID-19 pandemic, which has highlighted the importance of such work as an organisational concept and practice [8]. Briefly speaking, remote work "concerns any intellectual work carried out outside the normal place of work, whose effects are sent to the employer using information and communication technologies" [9]. Several studies have

been published which show increases in employee productivity [10] and organisational economic performance [11] due to remote work. One can also find studies that found a decrease in employee productivity due to RW and stopped offering this option to their employees [12,13].

In the initial period, remote work was treated as an employee's privilege or even a luxury [14]. Flexibility is no longer merely an additional asset but it is now also a competitive tool which organisations can use, for instance, to accomplish certain recruitment objectives or to gain a competitive edge [15–17].

In 2020, 12% of the EU employees who were aged 20–64 usually worked from home and in the past decade this percentage was at a steady level of about 5–6% [18,19]. Therefore, this type of work was not a popular practice [20,21]. Nonetheless, in response to the pandemic crisis and an unstable environment, one of the decisions was to universally switch to working from home [22].

In Poland, the percentage of people usually working from home in 2020 exhibited a nearly two-fold increase in relation to 2019 (4.6% vs. 8.9%) [23]. The COVID-19 pandemic and the subsequent period changed the attitude to remote work, which became a necessity for many organisations [24–27]. This arrangement is currently used by many enterprises, also with the aim of ensuring a proper work-life balance for their employees [28–31], improving the organisation's performance and reducing employee absenteeism [32,33]. The decision that was made by mid-level managers to adopt remote work as a form of employment was dictated furthermore by their convictions about the efficiency of the work that is performed by their employees and about information security measures having become more reliable [34].

Regarding remote work, many new, previously unknown problems have been faced by employers, such as: the organisation of the remote work, the supervision and monitoring of work performance, work efficiency, and employee support [35–38], as well as the intention to continue working remotely once the COVID-19 pandemic is over.

The aim of this paper is to analyse the influence of certain factors that were selected by the authors on the efficiency of remote work. The authors have assumed that the efficiency of performing work from home is the principal factor that is conditioning an assessment of remote work by enterprises. Another element to which special attention is paid is the intention to continue remote work in the future. This is related to assessing the efficiency of the work that is performed outside of the corporate office, but it can also be influenced by other factors. Based on the research results, the authors want to analyse the inclination of enterprises to continue remote work once the pandemic is over.

## 2. Supervision and Monitoring of Remote Work

Remote work performance has triggered the need to conduct ongoing monitoring [37,39] and to search for new measures of remote work efficiency. In the early 1980s, managers indicated trust and respect as the necessary attributes in their relationships with remote workers; they also pointed to the need for certain standards [40]. They, therefore, began searching for systems to support work monitoring, with a particular focus on performance monitoring [41]. However, it turned out that additional monitoring would be detrimental to remote workers [42] and could be replaced by information sharing and new forms of contact in order to obtain the expected results. Research has shown that the situational leadership model, which has been in use since the late 1960s in the United States, plays a major role here. It forms an appropriate tool which leaders can use to effectively influence their employees outside of the workplace with a view to improving their work performance [43]. However, as also revealed by research results, not every leadership style ensures the attainment of top performance levels, either among remote employees or among remote managers [44–46].

The monitoring of remote work is not always effective. Employees may deliberately delay responding to forms of monitoring or perform activities that are completely unrelated to their work [47]. To counteract such situations, employers make use of various forms

of supervision, as labour laws do not expressly define its limits [46–48]. The exercise of supervision is supported by numerous applications and systems that are intended for employee monitoring. The tools for remote work supervision can include the employee's filling in timesheets, adding notes and comments in files or generating reports to sum up their obtained results. There are also systems that can be used for detecting the lack of activity of a given user on a computer or recording time spent on social networking sites, as well as software tracking a user's location during work [49].

During the pandemic, traditional mechanisms of direct supervision are being exercised through digital platforms (ActivTrak, InterGuard, Veriato 360, Teramind, WorkSmart, Work Examiner and Sneek) that became popular, along with e-mails or phone calls, but these proved to be ineffective [49]. Inspection and supervision by managers are constantly taking on new forms. New procedures are being introduced and employees are required to produce written reports showing the extent of the work that they have performed during the day [50]. In addition, business intelligence and data analytics tools are often used to further monitor employees' work [49]. However, technologies for employee monitoring frequently require the installation of remote-control software, which can be a threat to employees and even perceived as "flexploitation" [51,52].

The positive influence of an assessment of work efficiency on reducing the scope of work supervision was confirmed by Wang et. al. These authors, in their research conclusions, did not explicitly demonstrate the desirable effect of additional monitoring on the efficiency of remote work [37]. Monitoring should be supported by motivating communication that is exchanged between managers and subordinated employees [53], along with the use of customised flexible work arrangements [54]. This will enable achieving both the work-life balance of the employee and better organisational performance [55,56].

## 3. Efficiency of Remote Work

Scientific literature demonstrates a need to establish measures for assessing work efficiency. Authors analyse work efficiency as viewed from an employer's perspective. In this context, benefits in the form of reduced labour costs and outlay which will be incurred when providing remote work should be considered. Savings are achieved due to there being no need to commute to work, the so-called office policy [57], the elimination of unproductive meetings [58], less sick leave and breaks [59] or, generally speaking, due to the lower costs of arranging the workplace [60–62]. This clearly suggests that the work efficiency measures are perceived in financial terms. However, consulting firms, in their reports, approach the efficiency assessment issue from a different angle [63,64]. It has been revealed that only one in five such companies in Poland has declared that its efficiency assessment was based on objective measures, such as key performance indicators (KPIs), and compared them to the results that were achieved before the pandemic [65]. Most companies, in turn, base their assessments on the opinions that are expressed by managers or employees, while one in three companies do not currently monitor the efficiency of their remote work. This stems from the shortage of appropriate tools and from insufficient home office equipment [66]. The growing inclination to supervise and monitor employees may trigger the need to grant consent to such supervision on the part of the employee [67].

The results of the questionnaire surveys that were conducted by D.P. Marasigan revealed a significant relationship between the efficiency of working from home and employee performance [57]. The author proved that employee performance during remote working was high but varied by gender and educational level.

Research results have been confirmed in practice—more specifically, in the reports of consultancy firms. Those that take into consideration the employer's perspective indicate that remote work contributes to reducing enterprise costs and to improving employee performance. The latter, however, decreases with the increase in the number of working hours and work intensity [68].

The efficiency of the remote work depends on the managers' ability to effectively engage and motivate their employees [69–71] and to influence changes in their work

patterns [72]. This requires both a substantial shift in the organisational culture towards outcome management and the establishment of relationships that are based on trust [73]. Regarding remote working in research, it has been found that increasing the usage of flexible work arrangements can improve productivity and creativity [74]. Increased productivity has resulted from the increased number of meetings that are held online and from the use of technologies of remote employee monitoring [75]. The recorded increase has frequently reached several percent [76,77]. The growth can be attributed to both the better use of working time (e.g., shorter breaks) and higher work efficiency [76]. Nonetheless, as revealed by research, the lack of adequate support for remote work has eventually resulted in the decreased efficiency of remote work [73,74,77,78].

To sum up, the research that has been conducted to date, which has not taken the pandemic period into consideration, is mostly positive about remote work, indicating not only increased work efficiency but also more effective working time, increased autonomy and employee independence [74,79]. However, more recent findings have revealed that the challenges that are related to remote work during the pandemic have a negative impact on the work efficiency as well as on the well-being of employees [37]. In consequence, some non-financial or financial support from an employer appears essential.

## 4. Remote Work Support

Organisations may offer financial support, additional office equipment, new software, free services, training, consulting, or additional non-wage benefits to those who are carrying out remote work [63]. There can also be other types of support that can influence work efficiency, such as social support, professional autonomy, monitoring of the workload, and an individual factor—namely, self-discipline [37]. Research highlights that the employer support should offer more autonomy and greater independence to the employees, which will increase their motivation to improve their work efficiency [80]. Moreover, the decreased dependency on co-workers' support results in increased motivation to act more independently [57,81,82]. Furthermore, research shows that this type of support contributes to increased efficiency, but this varies among employees and the positions in which they work [83]. A remote worker should be more productive when working from home, but this rule does not apply to all employees. Despite offering apparent freedom, employers implement additional objectives and use certain tools to foster productive work ethics. These include incentives for employees to self-evaluate and manage themselves with the support of digital technologies and the promotion of gamification [25,82–86].

Workforce fragmentation and the occurrence of employment inequalities of a social nature may constitute threats that may be faced by employers [69,84]. Organisations strive to mitigate these threats, this an aim which is reflected in their remote monitoring, management and supervision at the workplace. Such practices raise doubts regarding the maintenance of privacy and ethics in the process of contact with employees, as well as future security [87].

## 5. Intention to Continue Remote Work

The continuation of remote staff employment once the SARS-CoV-2 pandemic is over appears to be a major issue in the context of remote work. Howe et al. draws attention to the fact that the return to what used to be the pre-pandemic standard, i.e., working from a corporate office instead of the home office, can have some negative consequences. It can affect the morale of employees—those who prefer to work from home—resulting in reduced productivity and higher staff turnover [66]. As revealed by research, remote work can bring benefits to both parties, the employees and the employer. This work arrangement is often more suited to the younger generation and meets their expectations regarding work-life balance [88]. The benefits for the employer may result from, among other things, the reduced need for office space and lower costs of work tools [66]. At the same time, a number of disadvantages of flexible employment arrangements have been increasingly highlighted in the recent years. Soga et al. indicate that these can be grouped into several

general issues that are associated with health, socio-cultural, economic, spatial, technical and political factors [89].

In practice, companies that are operating in the same industries take very different approaches to remote work—from having all of their staff work from home, to having all of their staff work from the office, through various "in between" options. Althof et al. draw attention to the differing potentials of remote work depending on the type of work that is being performed—highly skilled employees of the business services sector have an increased potential to work remotely, more so than low-skilled employees of the service industry [90]. Team management in a remote work arrangement is more challenging when it comes to more complex and less clearly defined tasks, or to tasks that are implemented under varying conditions or related to new ventures [91]. Manko believes that the experience that is being gathered during the SARS-CoV-2 pandemic cannot be used as a direct source for predicting the future of remote work [91]. Other researchers indicate that, unless employees return to working from the office, managers will face some difficulties with adapting to permanent remote work and some tasks in this formula will be more challenging to perform, with large and resilient organisations coping the best [92]. At the same time, Radziukiewicz claims that the performance of work outside of the workplace will gradually become more common, with the hybrid model appearing the most likely solution, implying that work that is performed from the corporate office and outside of it will occur in specific cycles [93]. Research further indicates that this work arrangement is typically expected by employees [94].

## 6. Research Model and Hypotheses

In light of the above, the authors have proposed the research model that is presented in Figure 1. According to the authors, the factors that influence the efficiency of remote work include the attitude to remote work, the level of remote work control and the level of remote work support. An enterprise's attitude to the remote work exerts an additional impact on remote work control and on the level of remote work support. In turn, an assessment of the efficiency of the remote work, similar to remote work control and the employer's support, will have an impact on the intention to continue remote work in the future.

**Hypothesis 1 (H1).** *The better the enterprise's attitude to remote work, the higher the assessment of its efficiency.*

**Hypothesis 2 (H2).** *The better the enterprise's attitude to remote work, the lower the level of its control.*

**Hypothesis 3 (H3).** *The better the enterprise's attitude to remote work, the higher the level of its support.*

**Hypothesis 4 (H4).** *The higher the level of remote work control, the higher the assessment of its efficiency.*

**Hypothesis 5 (H5).** *The higher the level of remote work support, the higher the assessment of its efficiency.*

**Hypothesis 6 (H6).** *The higher the level of remote work control, the lower the intention to continue such work.*

**Hypothesis 7 (H7).** *The higher the level of remote work support, the higher the intention to continue such work.*

**Hypothesis 8 (H8).** *The higher the efficiency of remote work, the higher the intention to continue such work.*

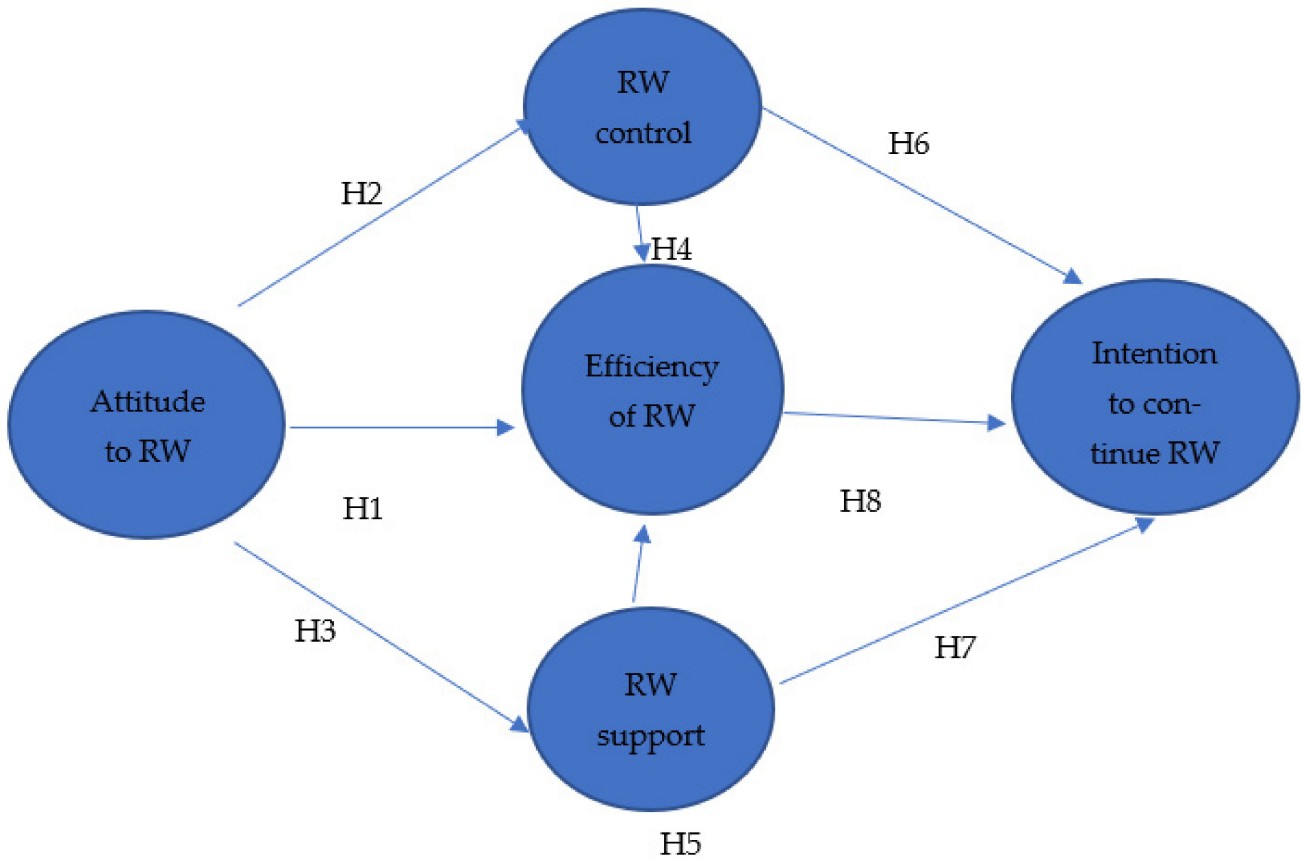

**Figure 1.** Research model.

## 7. Research Method

The research was conducted in May–June 2021 on Polish companies. Poland is a country that has been affected by SARS-CoV2, as have most European countries. A lockdown in Poland limiting economic activity was in force in periods from 13 March 2020, from 24 October 2020, and from 21 March 2021. These have concerned many Polish enterprises. Only in the second quarter of 2020, the decline in the GDP was 8% compared to the second quarter of 2019 (Podsumowanie lockdown-u w Polsce, Związek Przedsiębiorców i Pracodawców, Warszawa styczeń 2021, https://zpp.net.pl/wp-content/uploads/2021/01/25.01.20 21-Business-Paper-Podsumowanie-lockdownu-w-Polsce.pdf, accessed on 11 March 2022).

The research method that was used was a standardised questionnaire and the data were collected using CATI (computer assisted telephone interviewing). A 7-point Likert scale was used to answer the questions in the questionnaire. A pre-test was conducted with some employees in order to develop the readability of the questionnaire. Companies that used remote working during the pandemic were purposefully selected for the research sample. The respondents to the sample were the people with knowledge within the field of the remote work of the employees in a given company. Larger companies were included in the research sample in an over-representative manner, due to the fact that the data indicated that the majority of Polish enterprises are one-person businesses that do not employ any staff. Enterprises where only the owner works were excluded from the sample due to the purpose of the study.

A total of 256 enterprises were examined, of which 248 questionnaires were accepted for the analysis after verification. Enterprises with a differing number of employees were purposefully selected for the sample. The estimated percentages of the various enterprise sizes that are operating in Poland, as categorised by their number of employees, are [95]:

- from 1 to 9 employees—97%,
- 10–49 employees—2.2%,

- 50–249 employees—0.7%,
- 250 or more employees—0.2%

The characteristics of the research sample are presented in Table 1. The largest percentage of the surveyed enterprises—29.8%—were those that were employing between 10 and 49 people.

**Table 1.** Characteristics of the research sample (N = 248).

| Characteristic | Numbers | Percentages |
|---|---|---|
| Number of employees: | | |
| Up to 9 | 56 | 22.6 |
| 10–49 | 74 | 29.8 |
| 50–249 | 37 | 14.9 |
| 250 or more | 81 | 32.6 |
| Type of activity: | | |
| Production | 72 | 29.0 |
| Retail | 74 | 29.8 |
| Other | 102 | 41.1 |
| Annual turnover: * | | |
| Up to 224 thousand | 112 | 45.2 |
| From 224 to 22,000 thousand | 78 | 31.5 |
| From 22,000 to 111,000 thousand | 27 | 10.9 |
| From 111,000 to 222,000 thousand | 20 | 8.1 |
| Above 222,000 thousand | 11 | 4.4 |

* In EUR.

## 8. Model Calculation

In order to estimate the model and perform the analysis, partial least squares–structural equation modelling (PLS-SEM) was used. PLS-SEM allows one to analyse the relationships between latent variables, in the case of small research samples, and the non-normal distribution of variables [96]. The Shapiro–Wilk test showed that, for each of the observed variables, their distributions differ from the normal distribution. The data were analysed using the Smart PLS 3.3.7 program.

The first step in the analysis was to check the reliability and validity of the measurement. For this purpose, the values of Cronbach's alpha coefficients and composite reliability (CR) were calculated. The reliability of the internal consistency is satisfied if the indexes fall within the range of 0.7–0.95 [96]. The validity was assessed in two aspects: the convergent validity and discriminant validity. The convergent validity was assessed using the AVE (average variance extracted), the value of which should exceed 0.5. In the next step, the differential validity was examined by comparing the square root of the AVE with the appropriate correlation coefficients between latent variables in the model [97,98]. All of the variables in the model achieved the expected reliability and validity indicators (Table 2).

To measure multicollinearity, the variance inflation factor (VIF) of all the constructs was estimated. VIF values above 5 indicate strong collinearity among the indicators [99]. The calculated VIFs were below the threshold of 5. Using the PLS-SEM technique, we tested the hypotheses with the bootstrapping procedure, including 5000 trials. In this way, we were able to calculate the path coefficients, $p$ values, and $R^2$ values (Table 3).

**Table 2.** Reliability and validity of the constructs.

| | Cronbach's Alpha | Composite Reliability (CR) | Average Variance Extracted (AVE) | [1–5] |
|---|---|---|---|---|
| Attitude to remote work (ARW) [1] | 0.860 | 0.905 | 0.704 | 0.839 |
| Remote work control (RWC) [2] | 0.920 | 0.940 | 0.759 | **0.495** 0.871 |
| Remote work support (RWS) [3] | 0.772 | 0.862 | 0.676 | 0.757 **0.456** 0.822 |
| Efficiency of remote work (ERW) [4] | 0.904 | 0.933 | 0.777 | 0.660 0.496 **0.486** 0.882 |
| The intention to continue remote work (ICRW) [5] | 0.924 | 0.950 | 0.867 | 0.642 0.539 0.557 0.562 **0.931** |

**Table 3.** Hypothesis testing, path coefficient, R values, *p* values.

| Hypothesis | Linkages | Path Coefficients | *p*-Values | Hypothesis Testing |
|---|---|---|---|---|
| H1 | ARW → ERW | 0.603 | $p < 0.001$ | Supported |
| H2 | ARW → RWC | 0.495 | $p < 0.001$ | Supported |
| H3 | ARW → RWS | 0.757 | $p < 0.001$ | Supported |
| H4 | RWC → ERW | 0.229 | $p < 0.001$ | Supported |
| H5 | RWS → ERW | −0.074 | $p < 0.377$ | Rejected |
| H6 | RWC → ICRW | 0.262 | $p < 0.01$ | Supported |
| H7 | RWS → ICRW | 0.297 | $p < 0.001$ | Supported |
| H7 | ERW → ICRW | 0.288 | $p < 0.001$ | Supported |

This study has formulated 8 hypotheses, out of which three belong to the effects of three factors, the ARW on the ERW (H1), RWC (H2) and SRW (H3). The results show that, out of the three impacts, the effect of the ARW on the RWS is the strongest, where the path coefficient is 0.757, with a level of significance at $p < 0.001$. The effect of the ARW on the ERW score is weaker, as the path coefficient is 0.603, with a level of significance at $p < 0.001$. The ARW was the least affected by the RWC, as the path coefficient is 0.495, with a level of significance at $p < 0.001$. The ERW was also influenced by the RWC, while the influence of the RWS on the ERW is insignificant. The impact of the RWC level on the ERW is weaker (the path coefficient is 0.229, with a level of significance at $p < 0.001$) than that of the ARW on the ERW. The impacts of all of the analysed factors (ERW, RWC, and RWS) on the ICRW are similar. The influence of the RWS on the ICRW is slightly stronger than the other analysed factors—the path coefficient is 0.297, with a level of significance at $p < 0.001$. The assessment of the ERW showed a weaker influence on the ICRW (the path coefficient is 0.288, with a level of significance at $p < 0.001$), while the RWC has the weakest influence on the ICRW among the analysed factors (the path coefficient is 0.262, with a level of significance at $p < 0.01$).

$R^2$ was calculated as part of the next step. $R^2$ measures variance and the level of explanation for endogenous latent variables. It is a measure of the predictive power of a model [96]. The main factor that is explained in the model is the ERW. The $R^2$ value for this latent variable is 0.468, which is at the expected level. The model also explains the RWS ($R^2$-0.571) and ICRW ($R^2$-0.462) well. A slightly weaker prediction power applies to the RWC (0.242), which is also influenced by many other factors apart from those that were included in the research model. The validated model is presented in Figure 2.

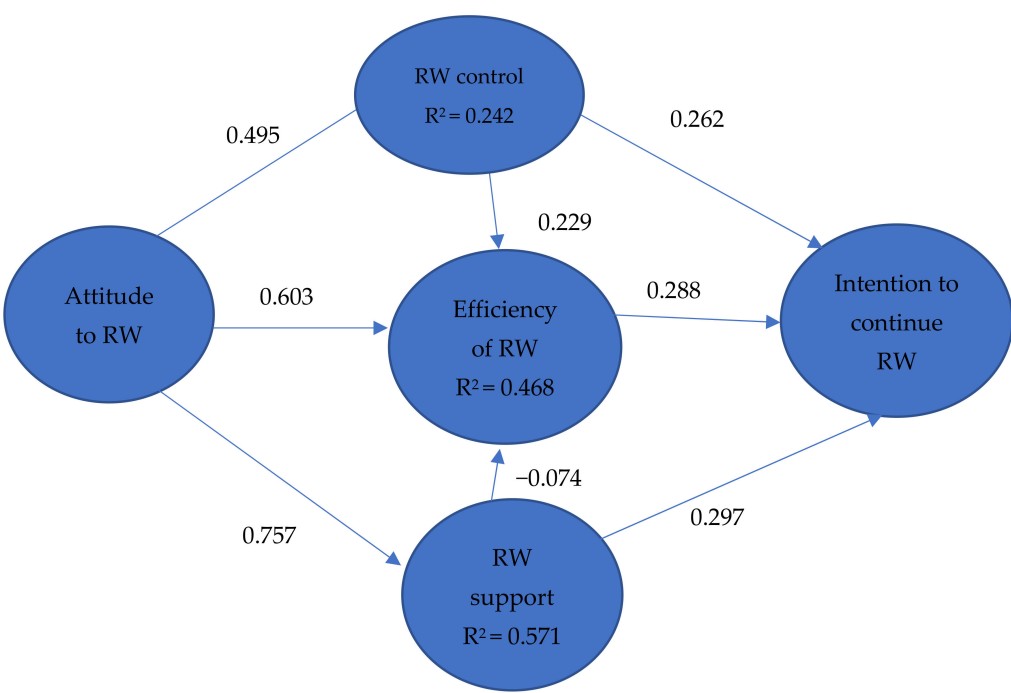

**Figure 2.** Model after validation.

## 9. Research Results

An enterprise's positive attitude to remote work exerts a positive impact on the efficiency of that remote work, the remote work control and the remote work support (which confirms H1, H2, and H3). The biggest impact was observed with regards to the remote work support. A better attitude to remote work influences, to the largest extent, an enterprise's support for performing work from remote locations outside of corporate offices. Among the enterprises that were surveyed, the following were most frequently indicated as elements of such support:

- Additional office equipment provided to the employee—31% of the enterprises;
- Remote work training—21% of the enterprises;
- Installation of additional computer programs—18% of the enterprises.

Financial support was declared by about 11% of the enterprises and it usually took the form of a remote work allowance or funds to cover additional costs (e.g., the Internet or equipment purchases).

In the relationships that are under analysis, and in the additional responses that were provided, there is an element of surprise and a necessary reaction to the compulsory switch to remote working. The attitude toward remote work, which was forced by the pandemic, produced reactions in the form of support for its performance by employees, most often working from home. A weaker relationship characterises the impact of the attitude to remote work on the assessment of its effectiveness, while the weakest relationship was found with the impact of the attitude to remote work on remote work control. Referring to these relationships, it can be concluded that remote work tends to be perceived as a temporary solution. Nearly 25% of the enterprises that were surveyed declared that they do not measure the efficiency of their employees who are working from home and 10.5% do not monitor these employees at all.

Remote work control exercises a positive impact on efficiency (which confirms H4), contrary to remote work support whose impact on efficiency has not been observed (which rejects H5). Therefore, the research conclusions that have been drawn, for instance by Wang et al., that indicate the lack of influence of remote work control on efficiency have not been confirmed in our research. It is most likely that an employer's perspective influences this relationship. Remote work control was usually exercised in the enterprises that were

surveyed by way of teleconferences (19% of the enterprises) and to-do lists (16.1%). The lack of the influence of remote work support on efficiency—which is contradictory to the findings that were published by Mustajab D. et al.—most likely results from the fact that such support is perceived in terms of additional expenses for remote work, which decreases the efficiency from an employer's perspective [77]. In turn, the benefits to which attention is drawn in the paper by Howe et al. probably did not occur in such a short-term period [66].

The intention to continue remote work is influenced by an assessment of its efficiency, the level of remote work support and the level of remote work control (which confirms H6–H8). The influence of these factors on the intention to continue remote work is similar. The most compelling impact of remote work support—and not of the assessment of its efficiency—on the intention to continue remote work may come as a surprise. It can be concluded that the greater the support that is provided for remote work by the enterprise, the higher its expectations that this form of work could be continued in the future. Indeed, 52% of the enterprises that were surveyed declared their willingness to continue remote work once the pandemic is over, but 61.3% of these claimed that its scope will be reduced. The impact of efficiency on the intention to continue working remotely once the pandemic is over, which turned out lower than the authors had expected, may indicate that some enterprises perceive remote work as an exceptional period, after which "normality will return"—this has most likely contributed to the obtained results.

Employers will be facing new challenges such as procrastination, ineffective communication, disruptions to working from home, and loneliness [37]. Ensuring job security and developing protection mechanisms against new cyber threats will also be essential [87].

## 10. Conclusions

In this new situation, with remote work no longer constituting a discretionary option but a more efficient (and economically interesting) alternative, it seems it is important to prepare for its skilful implementation. Some suggestions and challenges for managers, which they will need to tackle when implementing remote work into the management system of their organisation in the future, are outlined below.

Analysing work efficiency has become a challenge for employers under the new circumstances of remote work. This research shows that one-fourth of employers do not carry out such analyses and many others are unable to identify measures to be used to this end. The problem of measuring efficiency may be connected with the lack of appropriate tools, which were not needed before. Employers should adopt outcome or process indicators in order to verify the effects of the remote work [100]. Additionally, they should redesign work performance in the areas of remote work, as it requires a different arrangement of duties and tasks in order to increase efficiency [37,101,102]. The need to tailor flexible forms of work to individual employees will also prove to be quite challenging [103].

The extent of the support that is offered to remote workers creates a major challenge. It is predicted that employers will have to take on a much wider range of responsibilities in relation to this group of workers, from a wide array of training courses to provide support in such fields as work psychology. Based on our findings, the vast majority of employers (89%) do not provide their employees with additional financial support in connection with their remote work. This may stem from the conviction that the savings that are generated from there being no need to work from an office are greater than the additional expenses that are associated with performing remote work. Managers should find new ways to exercise their management using advanced information technologies, to communicate with their family, to plan tasks, to strive for increased efficiency [104], to develop innovative career paths, and to launch appropriate remote work support mechanisms [105,106].

Another challenge concerns the monitoring of the remote work, which entails a number of questions. Which tools should be used—those which monitor time and effort, or those which help to assess an employee's performance? Is trust in employees sufficient to foster their better performance? Or will poor monitoring be a source of procrastination? Re-

mote work creates new challenges for managers who need to cope with a different, probably not yet well-recognised management pattern. It, therefore, requires new managerial skills along with developing an outcome management style in an ICT environment. Managers should shift the focus in employee management from monitoring performance to performance management and place more emphasis on the work outcome rather than on the input [106]. This will, in turn, require the development of new performance management and assessment systems [25].

For those organisations which will opt for a hybrid work model, an additional challenge will be to shape the new organisational culture. They will need to strike a proper balance between a "tight" and "loose" organisational culture, referred to as tight–loose ambidexterity [107]. The new work model triggers the need to develop new internal procedures in the enterprise, as remote work requires a different arrangement of duties and tasks for the employees with whom there is no physical contact. As revealed by our research, less than 40% of the enterprises that were surveyed have implemented new rules and regulations concerning remote work.

## 11. Discussion

In summary, the experience that is being gathered during the pandemic period should contribute to a greater popularity of remote work in the future [95]. This form of work and, in particular, hybrid work are likely to become more widespread in organisations once the pandemic crisis is over [107]. Research shows that an employee can save between 28 and 50 working days per year which are wasted on commuting and that there are savings in office space [107,108]. However, this problem should be approached with caution, as providing remote work support will require a substantial investment in new technologies, mainly including IT tools (online whiteboards, software, high-class webcams, microphones, security, etc.), and in specialised training sessions and courses. Companies have already had to bear some of these expenses [109].

It can be expected that hybrid forms of telework will be widespread after the pandemic period, as indicated by the results of a Eurofound online survey that was conducted in July 2020. Over three-fourths of EU workers want to continue working from home at least sometimes in the future once the COVID-19 crisis is over, while only a few, (13%), would like to do so all the time. The majority, (78%), prefer a hybrid work model combining telework and remote work [69]. Employers may face a problem in decreasing employee commitment to work. Based on the research by Sull et al., one-fifth of human resource specialists claim that leaders have some doubts regarding the overall challenge of switching from office work to remote work and that the following issues have been highlighted: maintaining employee commitment (17%), efficiency (7%) and communication (5%) [110]. Heyns has been right to note that remote work remains a less popular practice compared to traditional forms of work, being dependent on the behavioural, cultural, and political aspects of the socio-technological changes and the interactions between them [103,111]. Remote work entails a number of security risks, including cyber threats [85,87,112], as well as a dissonance between the security and the privacy of employees [113]. Ensuring remote work security is conditional on shaping the employees' attitudes regarding their devotion to the employer organisation and their commitment to the work [83,109,114,115].

The contribution of the present article concerns taking into account the perspective of entrepreneurs regarding the problem of remote work and its effectiveness. The authors revealed that the company's attitude to remote work has the greatest impact on its support, and then the assessment of its effectiveness. In turn, the level of remote work support and the assessment of its effectiveness have the strongest impact on the intention to continue remote work in the future. In the opinion of enterprises, the control of remote work has a smaller impact on its effectiveness and the intention to continue. To the best of the authors' knowledge, the problem of the impact of the indicated factors on remote work has not been studied so far.

## 12. Limitations

The presented results, according to the authors, exhibit certain limitations. One of these is the pandemic period in which the survey was conducted. The survey sample was composed of enterprises using remote work arrangements during the pandemic and being forced to adapt to the limitations that were arising from such work. However, this period of limitation was finite in time and did not urge entrepreneurs to look at remote work from a different angle. For instance, the economies of scale, due to the reduced rented office space, could not be observed during this period. Another limitation is the selection of the survey sample in which large companies were overrepresented, which usually have fewer constraints, e.g., financial constraints related to providing remote work support. Therefore, this sample selection could have influenced the obtained results. A further limitation, according to the authors, is the research model itself, which took into consideration several basic elements influencing the use of remote work in enterprises. This model did not include, for example, the level of data security in electronic systems, which might be important for many enterprises. Most of the studies that have been conducted to date that deal with the issue of remote work during the pandemic period have focused on the remote employee's perspective. Approaching the problem from an employer's perspective might produce clearly distinct results, so this requires further investigation. Further research should concern the long-term impact of remote work on its effectiveness and other aspects affecting remote work from the perspective of enterprises, e.g., the perceived media richness.

**Author Contributions:** Conceptualisation: M.L., A.K. and Z.P.; Methodology: M.L. and Z.P.; Data analysis: M.L.; Original draft: Z.P., Manuscript editing: M.L., A.K. and Z.P.; Supervision: M.L., A.K. and Z.P. All authors have read and agreed to the published version of the manuscript.

**Funding:** This research received no external funding.

**Institutional Review Board Statement:** Not applicable.

**Informed Consent Statement:** Not applicable.

**Data Availability Statement:** Data availability on request.

**Conflicts of Interest:** The authors declare no conflict of interest.

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
