# Peer review of "Remote Work Efficiency from the Employers’ Perspective—What’s Next?"

_sustainability, doi:10.3390/su14074220_

Round 1

Reviewer 1 Report

The article under review can be evaluated positively in the current, perhaps post Covid era, brings new knowledge and is beneficial for science and research.
I recommend expanding the list of used scientific sources to include the knowledge of authors from the Central European area. These will contribute to higher scientific value:
- Father, RS. (2021). Reflections on actual situation of collective bargaining for the public servants and public services in Romania and in Europe. A theoretical and practical approach. JURIDICAL TRIBUNE-JURIDICA TRIBUNA 11 (2), pp.251-261
- Peracek, T. (2020). Human resources and their renumeration: managerial and legal backround. 13th International Scientific Conference on Reproduction of Human Capital - Mutual Links and Connection (RELIK) 2020 | RELIK 2020: REPRODUCTION OF HUMAN CAPITAL - MUTUAL LINKS AND CONNECTIONS, pp.454-465
- Adamišin, P., Butoracová Šindleryová, I. ÄŒajková, A. (2020).
CORONAVIRUS VS. REAL CAUSE OF THE EUROPEAN ECONOMIC CRISI– COMPARING SLOVAK AND GERMAN NATIONAL MODEL EXAMPLE, Online Journal Modeling the New Europe, 37, pp. 178-101, doi: 10.24193 / OJMNE.2021.37.05
- Srebalová, M. & Vojtech, F. (2021). SME Development in the Visegrad Area. Eurasian Studies in Business and Economics, 17, pp. 269–281, doi: 10.1007 / 978-3-030-65147-3_19

Author Response

Dear Reviewer

The authors would like to thank very much the reviewer for the contribution and positive evaluation of the article. We expanded the list of cited publications with selected ones suggested by the reviewer and other enriching the review of remote work research.

Reviewer 2 Report

  1. Various studies have already been conducted on workers' remote work due to the COVID-19 situation. There are many preceding studies similar to those suggested by the authors of this study based on the research results (proposals for efficient remote work). It is necessary to investigate and organize previous studies on similar topics. Please describe the difference between these studies and this study. In addition, please mention the contribution of this study in detail.

  1. This study was conducted based on a telephone survey conducted on workers working for Polish companies. What are the characteristics of Polish companies? It is necessary to provide specific reasons for conducting research on Polish companies.

  1. In the case of <Table 1>, I wonder what the criteria for classification for each item are. What is the basis for dividing the number of workers into 5 groups? I wonder why the industry in which the company belongs is divided into production, manufacturing, and other industries. In addition, the authors divided the year of establishment of the company as of 2015. Is 2015 the year when a major event occurs or becomes a turning point in Poland?

  1. The contents of the conclusion and discussion overlap, and the amount is too vast. It is necessary to delete overlapping parts and reduce the amount briefly. I think the paper should be organized more concisely to help readers read and understand the contents of the study.

  1. The limitations of the study also need to be summarized more briefly. It would be nice to present specific topics that can be conducted as follow-up studies.

  1. There are parts that do not fit the journal form. It is recommended to change the citation notation to [] rather than ().

I hope the opinions presented will help the constructive development of the paper.

Author Response

Firstly, the authors would like to thank the reviewer for his contribution and detailed evaluation of the article. We hope that the included changes to the text will improve its quality and meet the expectations of the reviewer.

  1. We investigated more texts and added them in the literature review. We emphasized the differences in our research compared to the previous ones. We mention the contribution of our study at the end of the conclusion part.
  2. Poland is a country affected by Sars-COV2 as are most European countries. Lockdown in Poland limiting economic activity was in force in the periods from March 27, 2020, from October 24, 2020, from March 21, 2021. They concerned many Polish enterprises. Only in the second quarter of 2020, the decline in GDP was 8% compared to the second quarter of 2019.
  3. Table 1 revises the classification of enterprises in terms of the number of employees into four groups. The classification of enterprises according to the industry they belong to is divided into three groups: production, retailing and others. The division is consistent with the classification of Statistics Poland. The classification of enterprises by year of establishment has been removed.
  4. Discussion and conclusions in the structure of the text have been combined and shortened.
  5. The limitation was completed in line with the reviewer's suggestions.
  6. Comments regarding the notation of citations will be taken into account after the comments of the technical editor.

The authors would like to thank you once again for your valuable comments.

Reviewer 3 Report

The manuscript deals with a topic in the mainstream, but it is well-built, and the contribution to the knowledge base is good. PLS-SEM is a popular method; the presentation of the results is detailed. The manuscript uses a similar style and structure to other papers based on this method. The used sources are acceptable. According to the structure, it is not usual that conclusion is followed by discussion. Please, check the opportunity to move it before the conclusions. I can recommend the publication after a minor revision, including a grammar refinement.

Author Response

The authors would like to thank the reviewer for the contribution and positive evaluation of the article. In line with the reviewers' comments, the conclusions and the discussion have been combined in one point.

Round 2

Reviewer 2 Report

The authors tried to reflect the comments suggested by the reviewer in the paper.

However, I would like you to clarify why the study should be conducted on Polish companies in the paper.

It seems necessary to ensure that the paper complies with the journal form.

Lastly, please check the spelling mistakes.

Author Response

We thank the Reviewer for the careful and insightful review of our manuscript. The comments will certainly help improve the article. In the research section, we added an explanation about conducting research in Poland. We also tried to correct language mistakes. 

Best regards

Authors